

# Response of bacterial community structure to different ecological niches and their functions in Korean pine forests

Rui-Qing Ji[1,2,*], Meng-Le Xie[1,3,*], Guan-Lin Li[1], Yang Xu[1], Ting-Ting Gao[1], Peng-Jie Xing[1], Li-Peng Meng[4] and Shu-Yan Liu[1]

[1] Engineering Research Center of Edible and Medicinal Fungi, Ministry of Education, Jilin Agricultural University, Changchun, Jilin Province, China
[2] Key Laboratory of Edible Fungus Resources Utilization in North China, Ministry of Agriculture and Rural Affairs, Jilin Agricultural University, Changchun, Jilin Province, China
[3] Life Science College, Northeast Normal University, Changchun, Jilin Province, China
[4] Wood Research Institute, Jilin Forestry Science Institute, Changchun, Jilin Province, China
[*] These authors contributed equally to this work.

Corresponding authors
Li-Peng Meng, Mxymlp@126.com, menglipengmlp@126.com
Shu-Yan Liu, liussyan@163.com

## ABSTRACT

A healthy plant microbiome is diverse, taxonomically-structured, and gives its plant host moderate advantages in growth, development, stress tolerance, and disease resistance. The plant microbiome varies with ecological niches and is influenced by variables that are complex and difficult to separate from each other, such as the plant species, soil, and environmental factors. To explore the composition, diversity, and functions of the bacterial community of Korean pine forests, we used high-throughput sequencing to study five areas with different forest ages from June to October 2017 in northeast China. We obtained 3,247 operational taxonomic units (OTUs) based on 16S rRNA gene sequencing via an Illumina Hi-seq platform. A total of 36 phyla and 159 known genera were classified. The Shannon index of the bacterial community from the rhizospheric soil was significantly higher ($p < 0.01$, $n = 10$) than that of the root tips. Beta-diversity analysis confirmed that the bacterial community of the rhizospheric soil was significantly different ($p < 0.001$) from the root tips. Nine bacterial phyla were dominant (relative richness > 1%) in the rhizospheric soil, but there were six dominant phyla in the root tips. Proteobacteria was the core flora in the root tips with a relative abundance of more than 50%. It is known that the formation of bacterial communities in the rhizospheric soil or the root is mainly caused by the processes of selection, and we found a relatively high abundance of a few dominant species. We further analyzed the correlations between the bacterial community from the rhizospheric soil with that of the root tips, as well as the correlations of the bacterial community with soil physicochemical properties and climate factors. We used Functional Annotation of the Prokaryotic Tax (FAPROTAX) to predict the functions of the bacterial community in the rhizospheric soil and root tips. Five related phototrophic functions, nine nitrogen cycle functions, two related chemoheterotrophic functions, and two others were predicted. The abundance of the bacteria phyla performing relevant functions was different in the rhizospheric soil than in the root tips. These functions were significantly influenced by the contents of nitrogen, phosphorus, and potassium in the soil habitat. The bacterial composition and functions in the rhizospheric soil and root tips of Korean

pine were analyzed, and the results demonstrated the importance of soil and plant species on the bacterial community in the below ground plant microbiome.

# INTRODUCTION

The root microbiome is one of the richest microbial ecosystems on earth, and bacteria are able to modulate plant growth and development (*Tringe et al., 2005*; *Hardoim, Van Overbeek & Elsas, 2008*; *Weyens et al., 2009*). Using high-throughput sequencing technology, we can develop a better understanding of bacterial community composition in plants across different ecological niches, including the rhizosphere, roots, and leaves (*Bulgarelli et al., 2013*; *Reinhold-Hurek et al., 2015*; *Müller et al., 2016*). Studies on *Arabidopsis thaliana* and some crop plants in which plant root exudates drive rhizospheric microbial composition have increased our understanding of the species composition of plant-associated microbial communities and their mechanisms (*Bodenhausen et al., 2014*; *Ofek-Lalzar et al., 2014*; *Schlaeppi et al., 2014*; *Bai et al., 2015*; *Bulgarelli et al., 2015*; *Edwards et al., 2015*; *Lebeis et al., 2015*; *De Souza et al., 2016*). The microbiome consists of fungi, bacteria, archaea, and oomycetes; possesses key ecological functions in forest ecological systems; and are usually well-adapted to a genotype of tree species (*Bonito et al., 2019*; *Hereira-Pacheco, Navarro-Noya & Dendooven, 2021*). Additionally, it is widely believed that climate factors and soil physicochemical characteristics significantly influence the soil microbiome, including soil organic matter content (*Fierer & Holden, 2003*), soil pH (*Chu et al., 2010*; *Hu et al., 2013*), air temperature (*Song et al., 2016*), and precipitation (*Wang et al., 2015*; *Wang et al., 2017a*; *Wang et al., 2017b*). We also understand that geographic distance and climate significantly shape the diversity patterns compared with soil organic matter from the large-scale biogeographical patterns of bacteria across forests (*Shay, Winder & Trofymow, 2015*; *Tu et al., 2016*; *Tian et al., 2017*; *Walters et al., 2018*). However, microbial distribution and assembly processes in micro-scale environments, such as root systems (rhizoplane and endosphere) and rhizospheric soils, are poorly understood. There is relatively little known regarding the properties of these host-associated communities (*Bulgarelli et al., 2013*; *Nicolitch et al., 2017*; *Koprivova et al., 2019*). Therefore, in this study, we selected pure or nearly pure Korean pine forests at different ages to determine their host-associated bacterial communities and dynamic changes.

The microbial community performs a wide variety of ecological functions in different niches, such as nitrogen and carbon cycling, decomposing organic matter, and promoting the growth of plants (*Kataoka & Futai, 2009*; *Wu et al., 2012*; *Johannes & Per, 2014*; *Kaiser et al., 2016*; *Wei et al., 2018*). Phototrophic and chemotrophic bacteria use organic and inorganic carbon sources for their growth and development. However, they also decompose and transform organic matter in the ecosystem that is difficult to decompose in the soil, constituting an important link in the carbon cycle. Simultaneously, bacteria are important

members involved in the nitrogen cycle because some bacteria can use atmospheric dinitrogen to provide the nitrogen source to most organisms that rely on bioavailable nitrogen for growth (*Kuypers, Marchant & Kartal, 2018*). Many functional prediction tools have been generated for both prokaryotic and eukaryotic microorganisms. For example, FUNGuild is a functional prediction tool for fungi that provides guild characteristics of the detected taxa (including saprotrophic, pathogenic, decomposer, or lichenivorous fungi) based on their taxonomic identity (*Nguyen et al., 2016*). A Phylogenetic Investigation of Communities by the Reconstruction of Unobserved States (PICRUSt) (*Langille et al., 2013*; *Douglas et al., 2020*) predicts the functional profiles from metagenomic 16S rRNA data (Tax4Fun) (*Aßhauer et al., 2015*; *Prada-Salcedo et al., 2020*), and Functional Annotation of the Prokaryotic Taxa (FAPROTAX) (*Louca, Parfrey & Doebeli, 2016*) was developed to predict bacterial and archaeal functions. FAPROTAX is the only tool that uses experimental data on culturable taxa to identify functional groups, metabolic phenotypes, or ecologically relevant functions (*Louca, Parfrey & Doebeli, 2016*). Furthermore, FAPROTAX can be used for the functional prediction of the biogeochemical cycle of environmental samples (*Louca, Parfrey & Doebeli, 2016*; *Varela et al., 2018*; *Amit et al., 2019*; *Deng et al., 2019*; *Sansupa et al., 2021*). Although FAPROTAX is unable to determine functional phenotypes of all taxa in the community, previous studies have shown that it was a helpful tool to predict functions related to biogeochemical dynamics (*Sansupa et al., 2021*), especially on the N and C cycles. For example, FAPROTAX was used to study the impact of microbial inoculation and fertilizer application on soil bacterial functions involved in the C and N cycles (*Wang et al., 2018*; *Gao et al., 2019*; *Li et al., 2019*). *Li et al. (2019)* revealed a significant effect of straw incorporation and nitrogen fertilization on hydrocarbon degradation and nitrogen fixation. Similarly, *Wang et al. (2018)* showed an increase of aerobic nitrite oxidation in soil inoculated with multi-species inoculants. Therefore, in this study, we selected FAPROTAX to predict bacteria functions in Korean pine root tips and rhizospheric soil.

Korean pine is a prized and ancient fossil tree species. Natural Korean pine forests are known as tertiary forests, and they have formed over hundreds of millions of years of replacement evolution (*Tian, 2011*). More than half of the world's Korean pines are distributed in northeast China, but their numbers have decreased with human activities, the expansion of farmland, forest area degradation, and environmental deterioration. Recently, there has been an increase in the cultivation of artificial Korean pine forests. The main issues facing Korean pine afforestation are the survival rate of seedlings and their resistance to pests and diseases (*Zhang et al., 2021*). However, due to unsuitable soil conditions, the survival rate of Korean pine trees is often very low (*Jiang, 2016*). Studies on the soil microbial composition of Korean pine forests provide a basis for selecting suitable woodland or seedling breeding bases. Ecological niches are polydimensional, and woody plants have adapted to both abiotic and biotic factors above and belowground by active or passive links. The fungal communities that inhabit the needles, buds, trunks, and branches of Korean pines change as the canopy and tree ages, and some fungal species can help the host increase its disease resistance (*Song & Huang, 2000*; *Song & Huang, 2001a*; *Song & Huang, 2001b*; *Song & Huang, 2001c*; *Song & Huang, 2001d*; *Song & Huang, 2001e*; *Song*

*& Huang, 2001f*). We know little about the natural microbial groups on the Korean pine root-base and how they change with different living conditions.

The objectives of this study were as follows: (1) to determine and compare the bacterial diversity and community structure of Korean pine forests across different ecological niches belowground; (2) to quantitatively assess the relative importance of multiple environmental variables (such as climate and soil physicochemical properties) in shaping bacterial diversity and community structure; (3) to quantitatively assess the relative importance of species correlations in root tips and rhizospheric soil to the community structure; and (4) to determine and compare the predicted functions across different ecological niches and their influencing factors.

## MATERIALS AND METHODS

### Study sites and experimental design

This study was conducted in the main distribution areas of Korean pine in the Changbai and Xiaoxing'anling Mountains of northeast China.

To determine the bacterial community structure, ecological functions, and factors influencing bacterial community structure and functions in the rhizospheric soil and root tips, we collected samples in August at five collection plots. We collected samples from June to October at Collection site 1 (CS1: coordinates 41°56′8″N, 126°30′12″E, altitude 600 m) and Collection site 2 (CS2: coordinates 41°59′28″N, 126°37′58″E, altitude 550 m), which were pure Korean pine artificial forests, approximately 60 years old. Collection site 3 (CS3: coordinates 42°01′45″N, 126°43′30″E, altitude 530 m) was a 5-year-old Korean pine seedling nursery (approximately 3,000 square meters). Collection site 4 (CS4: coordinates 42°22′55″N, 128°6′1″E, altitude 740 m) and Collection site 5 (CS5: coordinates 47°11′4″N, 128°52′52″E, altitude 340 m) were Korean pine natural forests >150-year-old, with other scattered trees and shrubs. The rhizospheric soil samples and root tip samples in these five sites are labeled in Table S1.

We could easily obtain the Korean pine root tips in the pure forests. However, to ensure that the root tip samples (which were mixed with other plants to varying degrees) originated from Korean pine in the natural forests, we collected root tip samples by digging along the root system exposed on the ground to the root tips. Typically, part of the Korean pine roots stick out of the ground (*Gao et al., 2020*). The rhizospheric soil samples were simultaneously collected at a depth of about 20 cm near the Korean pine roots.

### Sampling and processing of the root tips and rhizospheric soil

At CS3, we dug 10 saplings each from different locations and in the center of the seedbed until we collected about 50 saplings that we then pooled together into one root tip and one rhizospheric soil composite sample. At the other collection sites, we designed four plots (20 m × 20 m, including more than five Korean pine trees) in different locations. We selected five trees from each plot and collected the fine root segments (approximately 20–50 g) at four locations that were about 1 m from the trunk of tree (*Xing et al., 2020*) until we had 80 subsamples from the same collected site that we pooled into one sample. Finally, we had

10 root tip and 10 rhizosphere soil composite samples from the different sampling sites or at different sampling times for further processing.

Root tip samples from each collection site were placed in Ziplock bags and immediately maintained with dry ice. The rhizospheric soil came from the surface of the root tips. Root tip samples were soaked with sterile water and stirred with a glass stick, and the resulting soil suspension was centrifuged, its liquid discarded, and the precipitated soil was stored at −70 °C for further analysis (*Edwards et al., 2015*). Root tip samples were then soaked with a solution of 0.1% Tween 20 for one hour, and then the roots were thoroughly washed with tap water. The root tip samples were covered with absorbent tissue, dried, and crushed by hand for DNA extraction (*Fierer, Schimel & Holden, 2003*; *Bowsher et al., 2020*). The soil samples were air-dried at room temperature for determination of their physical and chemical properties.

## Rhizospheric soil physicochemical property analyses

The methods for analyzing the physicochemical properties of the rhizospheric soil samples followed *Bao (2000)*. Soil available phosphorus (SAP) was determined using the 0.5 mol/L NaHCO$_3$ extraction-molybdenum anti-colorimetric method. Soil organic matter (SOM) was detected using the potassium dichromate volumetric method. Soil effective nitrogen (SEN) was measured using the alkali diffusion method. Soil available potassium (SAK) was determined using 1 mol/L NH$_4$OAC extraction-flame photometry. Soil pH (SpH) was measured by potentiometry. The rainfall capacity (MR) and monthly mean air temperature (MT) during the collection process were obtained from the local weather bureau.

## DNA extraction and Illumina sequencing
### DNA extraction and PCR

The total community genomic DNA of the root tip or rhizospheric soil samples was extracted using a NucleoSpin® SoilIsolation Kit (MACHEREY-NAGEL, Germany) according to the manufacturer's instructions. The V3–V4 region of the 16S gene was amplified using modified versions of the primer set 338F (5′-ACTCCTACGGGAGGCAGCAG-3′) and 806 R (5′-GGACTACHVGGGTWTCTAAT-3′) (*Fu, Lv & Feng, 2016*). A PCR reaction system of 50 µL was established as follows: 30 ng genomic DNA, 4 µL PCR Primer Cocktail, 25 µL PCR Master Mix, and ddH$_2$O as needed. The PCR cycles were initiated using the following program: denaturation at 98 °C for 3 min, followed by 30 amplification cycles of 98 °C for 45 s, 55 °C for 45 s, 72 °C for 45 s, and a final extension of 72 °C for 7 min. The PCR products were purified with Ampure Xpbeads (Agencourt, Boston, MA, USA) to remove the unspecific products.

### Construction library

The qualified DNA samples were used to construct sequencing libraries. The final library was quantified in two different ways: by determining the average molecular length using an Agilent 2100 Bioanalyzer (Agilent DNA 1000 Reagents, Santa Clara, CA, USA), and quantifying the library using quantitative real-time PCR (qPCR) (EvaGreen$^{TM}$). The qualified libraries were sequenced pair-end on the system using the sequencing strategy PE250 (PE251 + 8 + 8 + 251) (Hi-Seq SBS Kit V2, Illumina, San Diego, CA, USA).

### Sequencing processing

To obtain clean reads, the raw data were filtered to eliminate the adapter pollution and low quality reads with USEARCH (v. 10.0.240) (*Edgar, 2013*) and UCHIME (v. 4.2.40) (*Edgar et al., 2011*). The paired-end reads with overlap were merged to tags. The tags were clustered to OTUs at 97% sequence similarity. The taxonomic ranks were assigned to OTU representative sequences using Ribosomal Database Project (RDP) Naïve Bayesian Classifier (v. 2.2) trained on the database (Greengene_2013_5_99) and RDP database (Release 11_5, 20160930), using 0.95 confidence values as cutoff. Finally, alpha diversity and the different species screening were analyzed based OTU and taxonomic ranks in R software (v. 3.1.1).

## Data processing and statistical analyses

Beta diversity analyses were conducted using QIIME (v. 1.80) (*Caporaso et al., 2010*) to evaluate the samples' differences in species complexity. Since there were differences in the sequencing depth across different samples, normalization was introduced. We used two methods to measure the beta diversity. The first was a principal coordinates analysis (PCoA) using the Bray–Curtis distance, which is an index that is commonly used to reflect the differences between two communities, and its value is between zero and one. A zero Bray–Curtis value represents an exactly similar community structure. Permutational multivariate analysis of variance (PERMANOVA) was used for the difference between the groups. The second method was a cluster tree based on the Bray–Curtis distance to illustrate the distance between the bacterial species, considering the abundance of sequences. A larger index indicates greater differences between the samples. The beta diversity heat map was drawn by 'aheatmap' in package 'NMF' of R software (v. 3.1.1).

The alpha diversity was applied to analyze the complexity of species diversity for each sample through several indices, including the Shannon and Simpson indices with QIIME (v. 1.80), and was displayed with R software (v. 3.1.1). The Shannon and Simpson values reflected the species diversity of the community affected by both richness and evenness of species, *i.e.,* the two values also considered the abundance of each species. The Simpson index was between 0 and 1, and the smaller the index value, the richer the diversity. We also used the Shannon index to explain the alpha diversity between the rhizospheric soil group and root tips group. The relative richness (RLR) of taxa was calculated to determine the dominant taxa (RLR > 1%) across different niches.

FAPROTAX (*Louca, Parfrey & Doebeli, 2016*) is more suitable at analyzing environmental samples based on 16S rDNA amplicon sequences. The database contains more than 7,600 functional annotations collected from more than 4,600 prokaryotic microorganisms in more than 80 functional groups (such as nitrate respiration, methane production, fermentation, and plant pathogens).

Moreover, Linear discriminant analysis Effect Size (LEfSe) determines the features (organisms, clades, or operational taxonomic units) most likely to explain differences between classes by coupling standard tests at a significance level of 0.05. We conducted the analyses using LEfSe to determine the biomarker taxa in different niches. Spearman's Correlation Coefficient (*Shannon et al., 2003*) was used to analyze the relationship between

biomarker families in root tips and all of the bacterial families of which the RLR was greater than 0.5% in their soil habitats.

The Spearman correlation analyses and redundancy analyses (RDA) were used to analyze the relationships between environmental factors and microbial diversity and composition (*Clarke et al., 2014*). The RDA and permutation test were conducted using the R (v. 3.1.1) program. $p < 0.05$ was considered statistically significant, and $p < 0.01$ was considered extremely statistically significant.

## RESULTS

### Bacterial composition in Korean pine root-based diversity measures
#### *Alpha diversity variations with ecological niches*

From June to October 2017, we generated approximately 424,170 sequences of the V3–V4 region of the 16S rRNA gene from 20 gross samples using an Illumina Hi-Seq sequencing platform. In total, we obtained 690,119 no-primer tags, and the average length was 420 bp. After the removal of the plant-derived and low-abundance OTUs, the high-quality reads were clustered into 3,247 OTUs using $\geq$ 97% sequence identity as the cut off. To compare the bacterial community diversity across all samples, the same survey effort level of 26,088 sequences (75% of the sequence number of the sample with the lowest sequence number) was selected. The final statistical analysis results after 100 iterations were obtained using comprehensive statistics.

When the Shannon indices of the rhizospheric soil group and root tip group were compared, the Shannon index of the rhizospheric soil group (6.35 ± 0.78) was significantly higher than that of the root tip group (3.56 ± 0.27) ($p < 0.01$, *n* value = 10), indicating that the bacterial species diversity of the rhizospheric soil was higher than that of the root tips (Fig. 1).

#### *Beta diversity of the rhizospheric soil and root tips*

The beta diversity was examined based on Bray–Curtis diversity distance using the principal coordinates analysis (PCoA). The first two coordinates of PCoA explained 77.17% of the variance in data (PC1 = 67.98%, PC2 = 9.19%) for all the samples (Fig. 2A). The bacterial communities clustered into two groups by niche, the rhizospheric soil group and the root group. The results of permutational multivariate analysis of variance (PERMANOVA) by Adonis showed extremely significance difference ($p < 0.001$) of the bacterial community structure between the rhizospheric soil and root tips.

Using weighted Bray–Curtis diversity distance (Fig. 2B), we clustered twenty samples into different groups: the rhizospheric soil group, CS1 (including CS1-T1, CS1-T2, CS1-T3, CS1-T4, CS1-T5, and CS1-T) and CS2 (CS2-T) were clustered into one group, CS3-T was an independent group that was different from other groups, and CS4-T and CS5-T were clustered into one group. From the results, the bacterial community from the rhizospheric soil was clustered according to different sampling plots and different tree ages. The samples in the root tips group were not grouped according to their sampling sites or the different tree ages: CS-G5 was one single branch, CS-G3 was another single branch, CS1-G2 and CS4-G were clustered into a group, and others were clustered into a group.

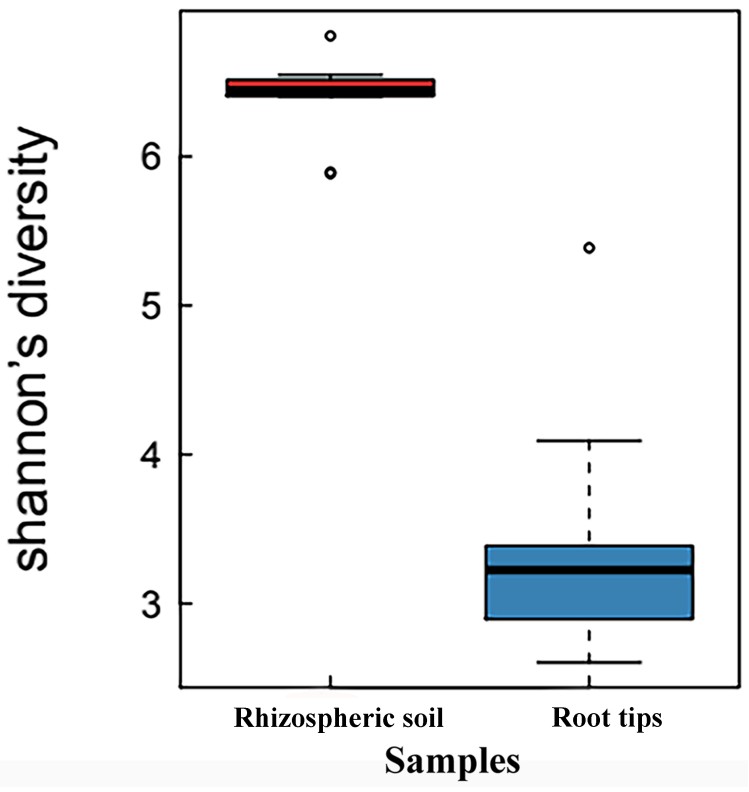

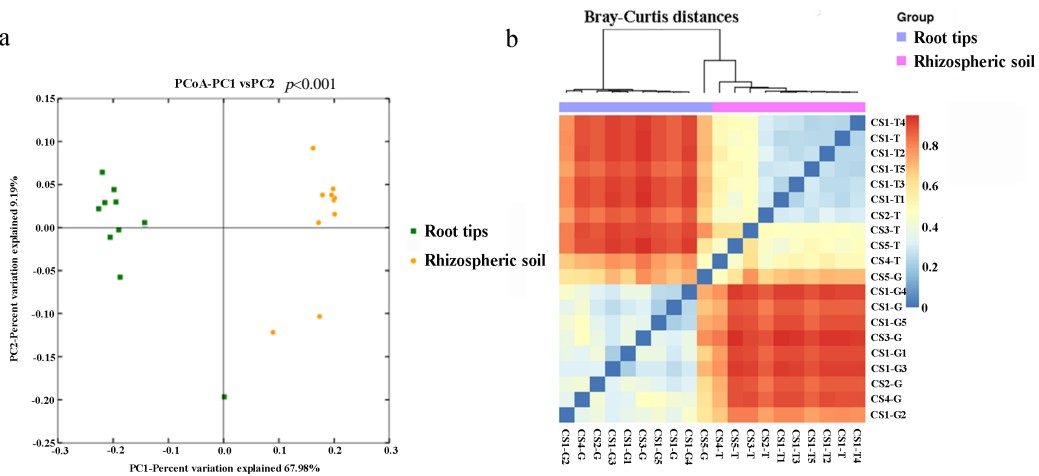

**Figure 1** Shannon index boxplot of bacterial communities between the rhizospheric soil and the root tips.

**Figure 2** Beta diversity responded to the bacterial community of the rhizospheric soil and the root tips. (A) Principal coordinates analysis (PCoA) based on Bray–Curits distance ($P < 0.001$, permutational multivariate analysis of variance (PERMANOVA) by Adonis); (B) heat map based on Bray–Curtis distance.

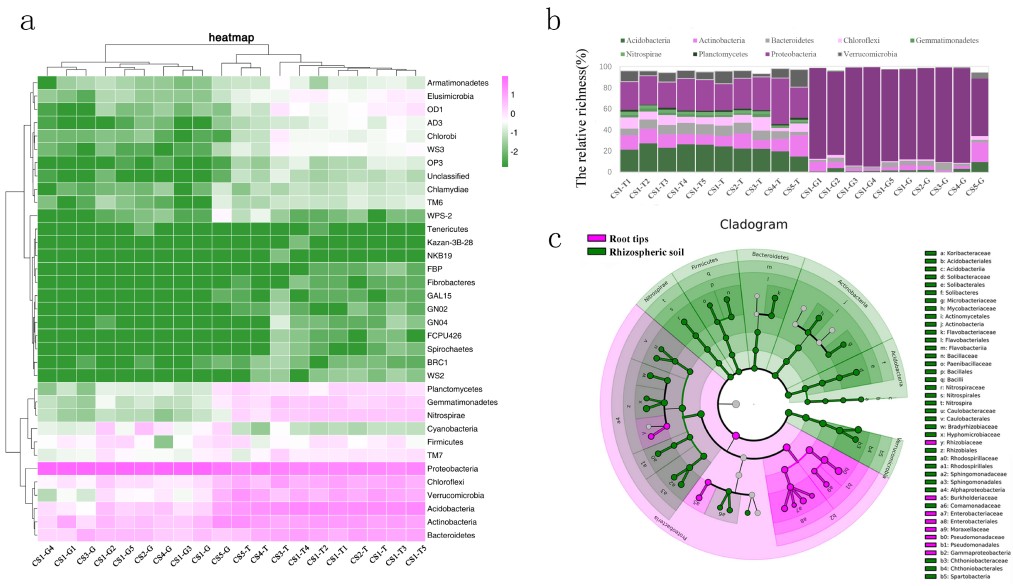

**Figure 3 The bacterial composition of the rhizospheric soil and the root tips based on taxonomic OTUs richness.** (A) Log-scaled percentage heat map of phylum-level; (B) the bacterial dominant phyla in different collection sites (the relative richness >0.01%); (C) the biomarker in root tips and in rhizospheric soil of Korean pine forest by LEfSe.

## Bacterial composition in the Korean pine roots based on taxonomic OTU richness

From the samples, we obtained 3,247 bacterial OTUs that belong to 35 known phyla and one unclassified taxon, 121 classes, 221 orders, 339 families, 159 genera, and 532 species (Table S2). The number of all taxonomic grades was higher in the samples from rhizospheric soil than those from root tips.

We compared the richness of different phyla by evaluating the logarithm with base 10 of RLR (Fig. 3A). From the collection site axis (the horizontal axis), two groups (rhizospheric soil and root tips) were completely separated according to the phyla RLR. The clustering of bacterial phyla from the root tips was irregular, while those from the rhizospheric soil were clustered according to forest lands. CS3-T was an independent group, CS4-T and CS5-T were clustered to the second group, and all of the rhizospheric soil samples from CS1 and CS2-T made up the third group.

From the phyla classification axis (the vertical axis), we clustered four general branches. The RLR of WPS2, BRC1, Spirochaetes, FCPU426, GN04, GN02, GAL15, NKB19, Kazan-3B-28, Tenericutes, Fibrobacteres, FBP, and WS2 displayed lowest in rhizospheric soil and root tips. The RLR of Proteobacteria, Bacteroidetes, Actinobacteria, Acidobacteria, Verrucomicrobia, and Chloroflexi were highest in the rhizospheric soil and root tips, and almost of them in the rhizospheric soil were higher than those in the root tips except the RLR of Proteobacteria. TM7, Firmicutes, Nitrospirae, Gemmatimonadetes, Planctomycetes, and Cyanobacteria had higher RLR in rhizospheric soil and root tips, and almost all were higher in the rhizospheric soil than in the root tips except Cyanobacteria. Although the RLR of

the TM6, Chlamydiae, Armatimonadetes, OD1, Elusimicrobia, WS3, Chlorobi, AD3, OP3, and unclassified taxa were lower in the rhizospheric soil and root tips, they were slightly higher in rhizospheric soil. (Fig. 3A).

Nine bacterial phyla (Acidobacteria, Actinobacteria, Bacteroidetes, Chloroflexi, Gemmatimonadetes, Nitrospirae, Planctomycetes, Proteobacteria, and Verrucomicrobia) were the dominant phyla in the rhizospheric soil, and six bacterial phyla (Acidobacteria, Actinobacteria, Bacteroidetes, Chloroflexi, Proteobacteria, and Verrucomicrobia) were the dominant phyla in the root tips. (Fig. 3B). We compared the bacterial community biomarkers of rhizospheric soil and root tips using a cladogram. The root tip biomarkers mostly belonged to Proteobacteria, including Gammaproteobacteria, Pseudomonadales, Psudomonadaceae, Moraxellaceae, Enterobacteriaceae, Enterobacteriales, Burkholderiaceae, and Rhizobiaceae. The rhizospheric soil biomarkers belonged to Nitrospirae, Firmicutes, Bacterioidetes, Actinnobacteria, Acidobacteria, and Verrucomicrobia, and included six classes, six orders, and 10 families (Fig. 3C).

## Bacterial composition of Korean pine roots based on the predicted ecological functions

We used FAPROTAX to predict the ecological functions of the bacterial community in the rhizospheric soil and root tip samples. We divided 18 related functions mainly into three categories: functions related to phototrophy, functions related to the nitrogen cycle, and functions related to chemoheterotrophy. The RLR of the related chemoheterotrophic functions (30.33%–81.19%) in the root tips was much higher than those related to the nitrogen cycle or phototrophy. In the rhizospheric soil, the relative abundances of the three functions were uniform, while the relative abundance of those related to chemoheterotrophic functions and nitrogen cycle was slightly higher than those related to phototrophic functions.

We used cluster analysis of the logarithm with base 10 of the RLR of ecological functions and samples (Fig. 4). From the collection site axis (the horizontal axis), three groups (rhizospheric soil and root tips) were completely separated, all of the rhizospheric soil was clustered into one group, the root tip samples of CS2-G and CS5-G were clustered into one group, and the rest of the root tips were clustered into another group. From the function axis (the vertical axis), two big groups were completely separated. In the first big group, the chemoheterotrophic and aerobic chemoheterotrophic RLR was higher in the rhizospheric soil and root tips, which were clustered into one subgroup. The other subgroup showed the functions of nitrate respiration, nitrate reduction, nitrogen respiration, and nitrite respiration, and the RLR from the rhizospheric soil was higher than that of the root tips. Fermentation and symbionts or animal parasites clustered into another subgroup, which showed lower RLR in the rhizospheric soil than in the root tips. In the other big group, the function of nitrogen fixation stood alone as a subgroup, and the others were in another subgroup that included phototrophic (anoxygenic photoautotrophy sulfur oxidizing, anoxygenic photoautotrophy, photoautotrophy, photoheterotrophy, and phototrophy) and some nitrogen cycle (aerobic ammonia oxidation, aerobic nitrite

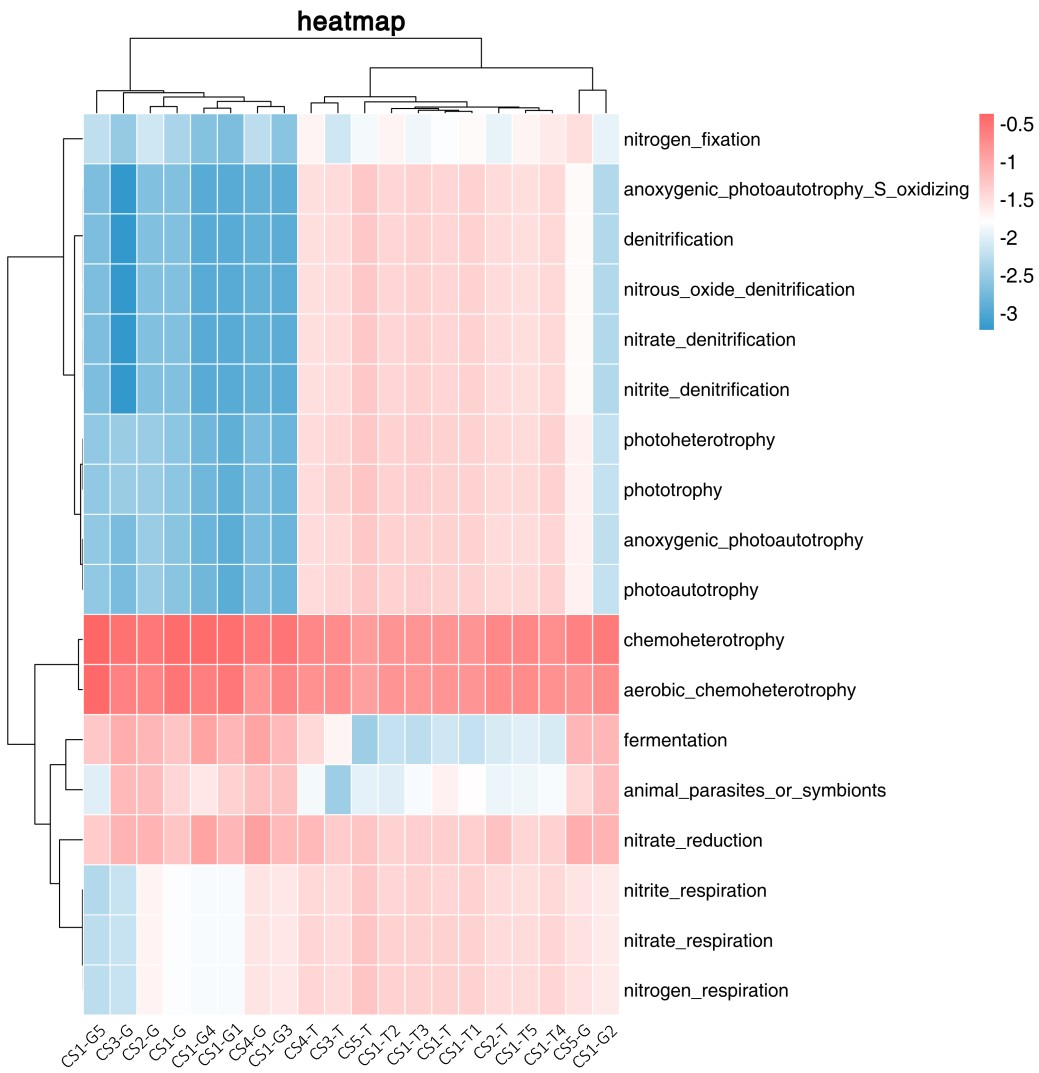

**Figure 4** Log-scaled percentage heatmap of bacterial functions predicted in different samples.

oxidation, nitrification, nitrate denitrification, nitrite denitrification, nitrous oxide denitrification, nitrate ammonification, and nitrite ammonification) functions.

## Climate factors, soil physicochemical properties, and the impacts on the bacterial community structure

### Climate factors and soil physicochemical properties

The climate factors and soil physical and chemical properties across the five sampling sites are shown in Tables 1A & 1B. Between June and October, almost all physicochemical properties changed, except the SpH. The SEN was highest in August, SAK was significantly highest in June, and SAP and SOM were the highest in September. MR and MT were the highest in August. Based on the Spearman correlation analyses of these factors (Figs. 5A & 5B), there were significant positive correlations between a number of factors, including the

**Table 1    Soil physical and chemical properties.**

| (a) Collection site 1 from June to October 2017. | | | | | | |
|---|---|---|---|---|---|---|
| **Sampling time at CS1** | **pH** | **SEN (mg kg⁻¹)** | **SAK (mg kg⁻¹)** | **SAP (mg kg⁻¹)** | **SOM (g kg⁻¹)** | **MT (°C)** | **MR (mm)** |
| June | $6.29 \pm 0.07$ | $182.53 \pm 6.88$ | $295.79 \pm 7.49$ | $2.12 \pm 0.15$ | $115.22 \pm 1.60$ | 17.4 | 74.8 |
| July | $6.49 \pm 0.03$ | $216.63 \pm 18.61$ | $259.82 \pm 5.66$ | $2.28 \pm 0.61$ | $69.12 \pm 1.42$ | 22.7 | 172.2 |
| August | $6.37 \pm 0.08$ | $247.62 \pm 28.38$ | $251.64 \pm 2.83$ | $1.21 \pm 0.27$ | $101.31 \pm 3.15$ | 23.9 | 317.0 |
| September | $6.29 \pm 0.12$ | $218.01 \pm 1.79$ | $261.45 \pm 10.21$ | $3.57 \pm 1.06$ | $134.05 \pm 2.54$ | 15.0 | 78.6 |
| October | $6.30 \pm 0.12$ | $214.56 \pm 5.20$ | $263.09 \pm 9.81$ | $1.52 \pm 0.79$ | $112.07 \pm 6.25$ | 7.8 | 70.2 |

| (b) Different collection sites in August 2017. | | | | | | |
|---|---|---|---|---|---|---|
| **Sampling sites** | **pH** | **SEN (mg kg⁻¹)** | **SAK (mg kg⁻¹)** | **SAP (mg kg⁻¹)** | **SOM (g kg⁻¹)** | **MT (°C)** | **MR (mm)** |
| CS1 | $6.37 \pm 0.08$ | $247.62 \pm 28.38$ | $251.64 \pm 2.83$ | $1.21 \pm 0.27$ | $101.30 \pm 3.15$ | 23.90 | 317.00 |
| CS2 | $6.14 \pm 0.05$ | $230.06 \pm 4.30$ | $281.08 \pm 7.49$ | $7.77 \pm 0.23$ | $118.12 \pm 3.12$ | 21.40 | 373.80 |
| CS3 | $6.88 \pm 0.29$ | $231.78 \pm 4.66$ | $181.33 \pm 7.49$ | $8.50 \pm 3.92$ | $138.42 \pm 8.36$ | 21.30 | 336.90 |
| CS4 | $5.40 \pm 0.17$ | $372.99 \pm 10.93$ | $498.57 \pm 4.91$ | $6.98 \pm 1.83$ | $230.53 \pm 4.91$ | 19.60 | 297.40 |
| CS5 | $5.66 \pm 0.26$ | $321.67 \pm 4.66$ | $303.97 \pm 2.83$ | $8.04 \pm 1.39$ | $185.39 \pm 3.71$ | 18.40 | 186.00 |

**Notes.**
The physical and chemical properties of soil were represented by the mean $\pm$ SD, $n$ value = 3.

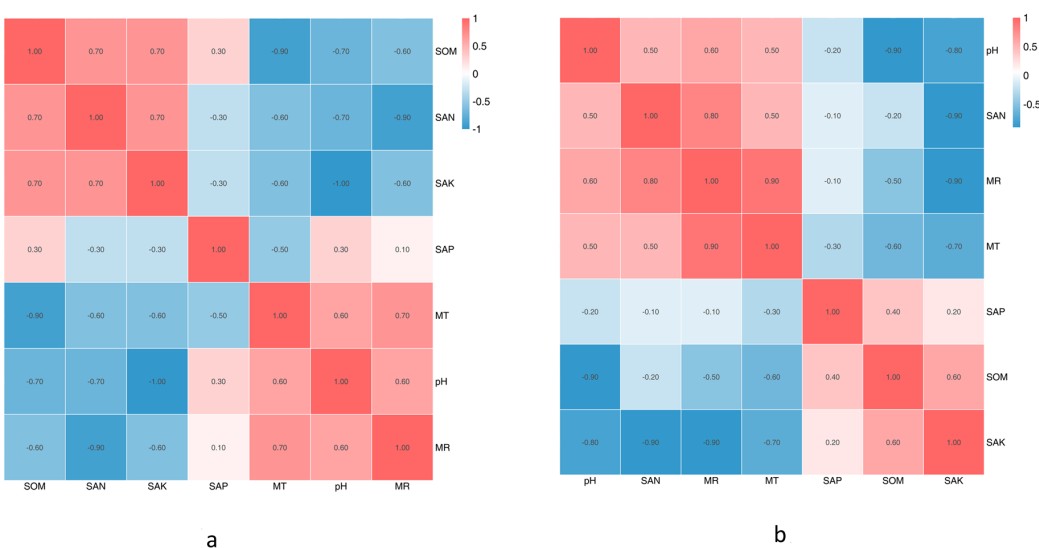

**Figure 5    The correlation of soil physical and chemical properties.** (A) Results in CS1 from June to October; (B) Results in different sites.

contents of SAK and MR (Pr = 0.9, Sig. = 0.037), SAK and SEN (Pr = 0.9, Sig. = 0.037), SOM and SEN (Pr = 0.9, Sig. = 0.037), SOM and SpH (Pr = 0.9, Sig. = 0.037), MR and SAK (Pr = 0.9, Sig. = 0.037), and MR and MT (Pr = 0.9, Sig. = 0.037). There was no significant correlation among the other factors.

The SpH at CS3 was higher than that of other collection sites. The SpH declined with forest age, while SEN, SAP, and SOM increased (Table 1B). The SAP was lowest at CS1, while the others remained stable. At CS4, it exhibited the lowest pH and the significantly

highest SEN, SAK, SAP, and SOM. The climate factors (mean air temperature and rainfall) were different across different collection sites. Based on the Spearman correlation analyses among these factors, there were opposite correlations between SpH and SAK (Pr = −1.0, Sig. = 0.037), MT and SOM (Pr = −1.0, Sig. = 0.037), and MR and SEN (Pr = −0.9, Sig. = 0.037).

### Effects of soil physicochemical properties on bacterial community structure

Samples collected from CS1 at different months showed that the SEN (Pr = 0.9, Sig. = 0.037) and SOM (Pr = 0.9, Sig. = 0.037) had significant correlations with the Shannon index of rhizospheric soil bacteria. The SpH had a significantly positive correlation ($p = 0.03$) with the RLR of the dominant bacteria, and as SpH increased, the RLR of Actinobacteria, Acidobacteria, and Nitrospirae increased (Fig. 6A). The SOM also had certain influence ($p = 0.08$) on the dominant bacteria RLR (Fig. 6B).

In different sampling sites at different forest ages, the Shannon index of rhizospheric soil bacteria was impacted by the SpH (Pr = 0.9, Sig. = 0.037), SEN (Pr = −0.9, Sig. = 0.037), and SAK (Pr = −0.9, Sig. = 0.037). The SOM had a significantly positive correlation ($p = 0.03$) with the dominant bacteria RLR (Figs. 6C & 6D).

## DISCUSSION

### Environmental factors determine the distribution of rhizospheric soil bacteria from Korean pine

Although our results showed that the rhizospheric bacterial composition changed regularly with tree age and collection plots (Figs. 2 & 3A), and that soil physical and chemical properties and climate factors were the factors affecting bacterial community structure, we could not draw conclusions due to sampling size limitations. The distribution of bacteria across different niches was driven by varying factors. In rhizospheric soil, the distribution of bacteria was correlated with the habitat, physicochemical characteristics of the habitat soil, and seasonal factors. Other studies have shown that SpH and SOM drive soil microbial communities at a global scale or in a specific region (*Lauber et al., 2009*; *Rousk et al., 2010*; *Bahram et al., 2018*), which confirmed our hypothesis that physicochemical properties are the most important factors affecting rhizospheric soil microbial composition and changes in the RLR of the rhizospheric soil bacteria across seasons and collection sites. For example, the RLR of Chloroflexi and Verrucomicrobia at CS3-T had positive and negative correlations with high SpH and low SEN and SAK, but we found no direct evidence for the influence of climate factors on bacterial communities. Perhaps they play an indirect role where they could influence the SpH, SAK, and SEN. At a regional scale, differences in climatic variables are quite small, which could be due to soil physical and chemical factors being the dominant factors (*Bahram et al., 2018*; *Sun, 2020*).

According to our bacterial species composition and dominant phyla results, the distribution of rhizospheric soil bacteria from Korean pine forests lands was not uniform. A few dominant species and a large number of rare species were shown. This pattern is very similar across all groups of organisms in different environmental conditions (*Wang et al., 2017a*; *Wang et al., 2017b*). Acidobacteria and Proteobacteria were the predominant

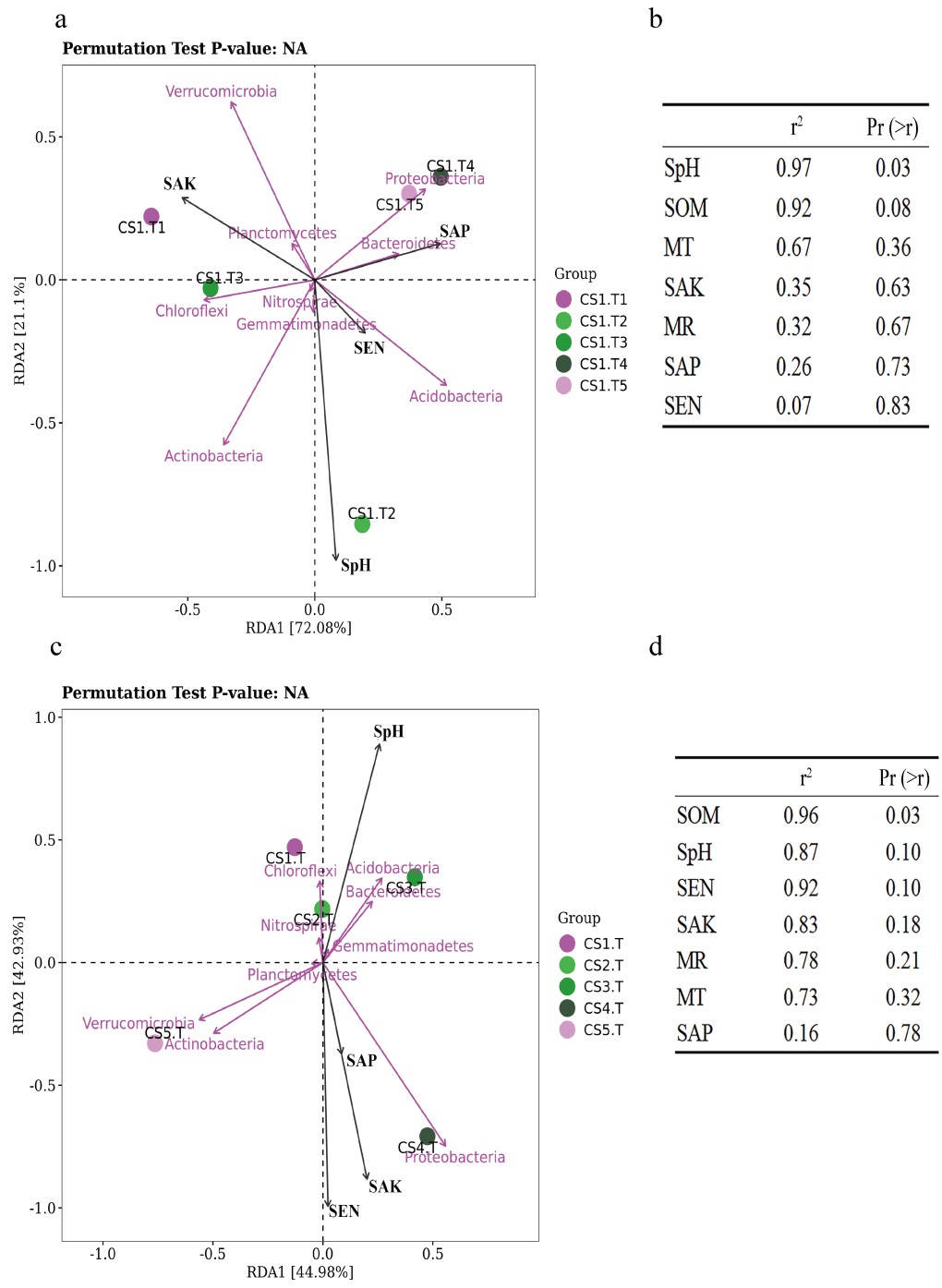

**Figure 6** **The RDA results of the dominant soil bacteria phyla in relation to soil physical and chemical factors.** (A) Results in CS1 from June to October 2017 and (B) significance analyses; (C) results in different sites and (D) significance analyses. The red arrows represent dominant bacterial phyla. The blue arrows represent soil chemical and physical characteristics and climate factors: SpH, MR, MT, SAN, SAP, SAK, SOM. Significant code: an asterisk (*) means $Pr < 0.05$ level.

phyla in the bacterial communities of other typical forests (*Lauber et al., 2009*; *Nemergut et al., 2010*; *Madigan et al., 2012*; *Lee & Eo, 2018*), as well as in grasslands and agricultural crop soils (*Will et al., 2010*; *Bulgarelli et al., 2012*) even though the soil conditions are very different. For example, forest soil has a higher accumulation of carbon (*Blaško et al., 2020*) and higher amounts of root exudates (*Grayston, Vaughan & Jones, 1997*). This may be due to the high morphological and metabolic diversity of Proteobacteria (*Kersters et al., 2006*) or the fact that Proteobacteria and Acidobacteria encode the genomic characteristics of high-affinity ATP transporters that cause them to be contained in various proteins (*Ward et al., 2009*). Therefore, the selection process determines rhizospheric soil bacteria community building in Korean pine forests, not dispersal limitations, which confirms the theory that environmental factors determine the distribution of microbial community structure, first proposed by *Baas-Beeking (1934)*.

## Root microbiota distribution of Korean pine is determined by the host's selection process

Plant species, rhizosphere sediments, and microbial interactions are the known determinants of root-related microbiota that are distinct from the rhizospheric soil microbiota (*Thiergart et al., 2020*). Numerous studies have found that plants can selectively recruit microbes from the soil to establish a characteristic microbiota on their roots (*Berg & Smalla, 2009*; *Bulgarelli et al., 2012*; *Edwards et al., 2015*; *Lebeis et al., 2015*; *Niu et al., 2017*; *Schlemper et al., 2017*; *Stringlis et al., 2018*; *Zhalnina et al., 2018*). The bacterial phyla Proteobacteria, Firmicutes, Bacteroidetes, and Actinobacteria often make up the bacterial diversity of root microbiota (*Bednarek et al., 2005*; *Van de Mortel et al., 2012*; *Kwon et al., 2016*; *Almario et al., 2017*; *Castrillo et al., 2017*; *Zhalnina et al., 2018*). Our hypothesis that the root system of Korean pine must have its own stable microbial composition during its long life that does not change with the seasons, soil physicochemical parameters, or its age, was confirmed to some extent by our study. According to our results, the bacterial community from the Korean pine root tip samples was dominated by Proteobacteria (approximately 50%), creating a stable bacterial community when compared with rhizospheric soil (Figs. 2 and 3). Other phyla, such as Actinobacteria, Acidobacteria, Bacteroidetes, Chloroflexi, and Verrucomicrobia, made up a smaller amount of each of the communities. The bacteria community in the Korean pine root tips were mainly the result of selection because the typical community pattern was a few dominant species and a large number of rare species.

## The bacterial community in rhizospheric soil impacts the bacterial composition of root tips

Soil bacterial composition has the most influence on root microbial composition (*Walters et al., 2018*) and is also the bacterial reservoir for above-ground plant microbiota (*Zarraonaindia et al., 2015*). In different ecological districts, multiple network relationships were found among microbial species (*Xiong et al., 2018*; *Yurgel et al., 2018*). These views were confirmed by our results. Based on the results of our bacterial compositions in the rhizospheric soil and root tips, and combined with previous research, we analyzed the results of continuous sampling from June to October in the same sample site. More than
a  b

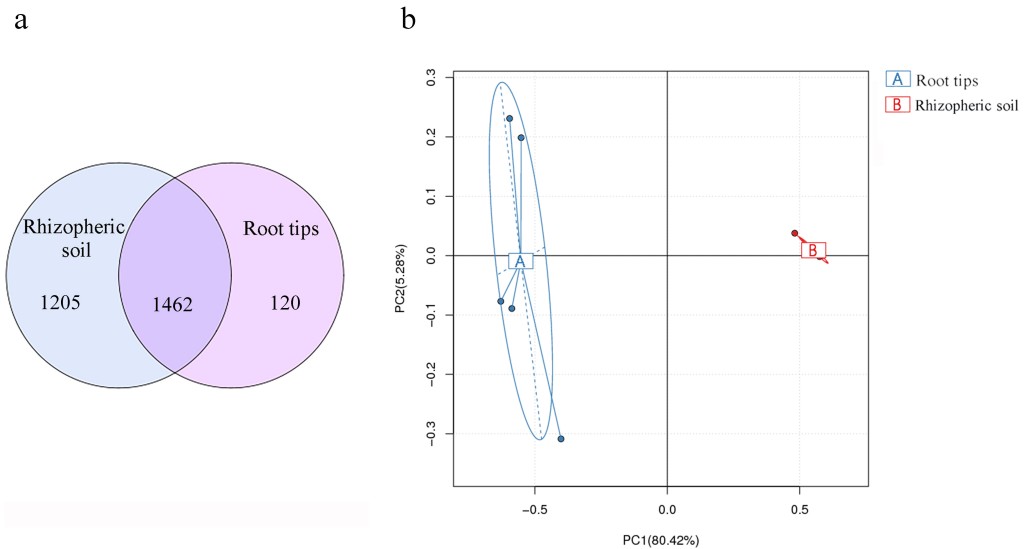

**Figure 7  Bacterial community differences between the root tips and the rhizospheric soil.** (A) Venn diagram representation of shared OTUs across different samples or groups; (B) PCA based on OTUs abundance.

90% of OTUs of the root tip bacteria were shared with those in the rhizospheric soil habitat (Fig. 7A), although the richness differed according to the PCA between the root tip and the rhizospheric soil bacteria (Fig. 7B). We concluded that the bacterial composition of root tips originates from rhizospheric soil.

Mutualism, neutralism, competition, parasitism, and predation determine the composition and function of the microbial community in the ecosystem (*Zengler & Zaramela, 2018*). *Gralka et al. (2020)* stressed that trophic interactions are the drivers of microbial community assembly. Our results showed that the biomarkers of families in roots had some correlations with bacteria from the root itself and the rhizospheric soil based on the RLR of families, such as Enterobacteriacea, which displayed an extremely significant negative correlation with family OC28 ($r = -0.950$, $p = 0.007$), and Pseudomonadaceae, which showed an extremely significant negative correlation with Nocardioidaceae ($r = -0.949$, $p = 0.007$) (Tables S4A & S4B). Based on our current knowledge, we cannot fully understand the relationship between different niche microorganisms, but the RLR relationship will lay the foundation for further research in these areas. A recent paper reported that the life cycle of the arbuscular mycorrhizal fungus *Rhizophagus clarus* could produce spores and complete its life cycle under the conditions of co-culture with *Paenibacillus validus* (*Sachiko et al., 2021*), which verifies the interaction between species and lays a foundation for the development of more beneficial microorganisms.

## Bacteria functions predicted in the root tips differ from those in rhizospheric soil

We found that the functions of bacteria are also mainly separately clustered in the rhizospheric soil and root tips, which is consistent with bacterial species composition

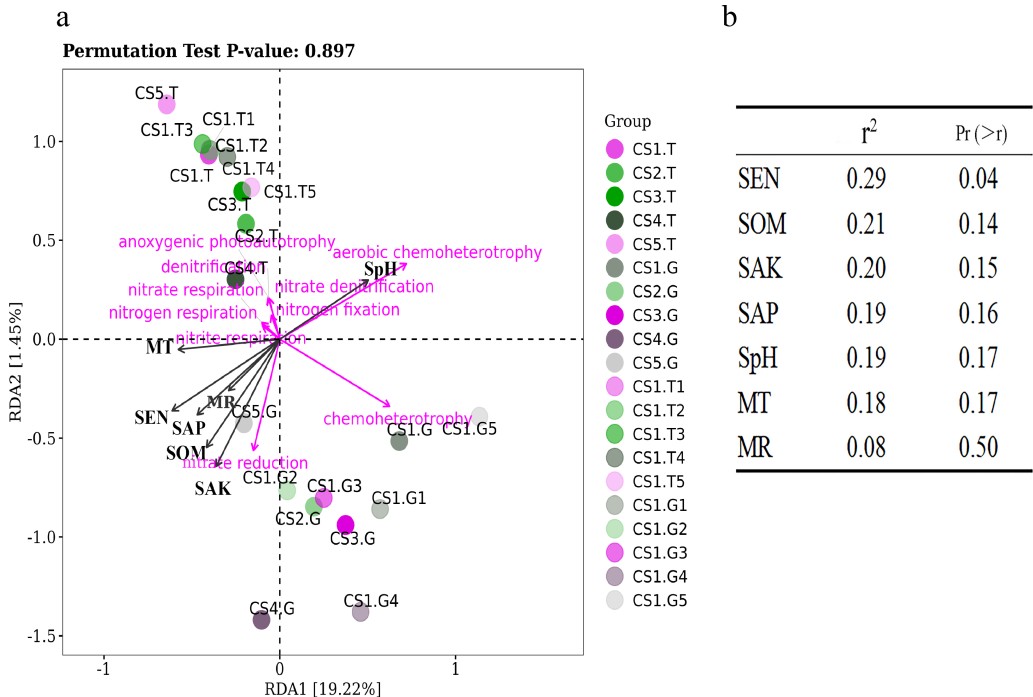

**Figure 8 The RDA results of bacteria functions predicted in relation to environmental factors of sample sites.** (A) RDA results of all samples and (B) Significance analyses.

(Figs. 3A & 4). This confirmed that the bacterial functions across different niches were driven by different bacterial communities (*Falkowski, Fenchel & Delong, 2008*; *Bai et al., 2015*; *Louca, Parfrey & Doebeli, 2016*). The core phylum of Korean pine root tips is Proteobacteria, which is consistent with the abundance of chemoheterotrophy. Some bacteria genera, like *Burkholderia* and *Rhizobium*, belonged to Proteobacteria and are also known for covering 'basic' functions such as nitrogen cycling and organic matter decomposition (*Lladó, Lopez-Mondejar & Baldrian, 2018*; *Sobti, Arora & Kothari, 2019*). In our limited sample size, we found that the RLR of phototrophic, nitrogen, and chemoheterotrophic bacterial functions predicted in the rhizospheric soil were comparably stable with growing seasons and collection plots, but varied in the root tips. Of course, we need to further increase the sampling scale to verify these rules.

We have tried to reveal the correlations of function bacteria RLR with soil and climate factors using RDA (Fig. 8). The first two coordinates of RDA explained only 20.67% of the variance in data (RDA1 = 19.22%, PC2 = 1.45%), which illustrated that the ecological functions of bacteria were partially impacted by soil and climate factors under normal growing conditions (not extreme environment). Our results showed that the RLR of bacterial function groups in the rhizospheric soil was separated from the root tips by RDA (Fig. 8A), which remain consistent with the bacterial communities. Moreover, SEN had some positive correlations with the RLR of nitrite respiration, nitrate reduction, and

nitrogen respiration (Figs. 8A and 8B), perhaps because SEN takes part in the nitrogen cycle.

## CONCLUSION

In this study, we found nine dominant bacterial phyla in the rhizosphere soil of Korean pine (Proteobacteria, Acidobacteria, Actinobacteria, Bacteroidetes, Chloroflexi, Verrucomicrobia, Nitrospirae, Gemmatimonadetes, and Planctomycetes) and six dominant bacterial phyla in the root tips (Proteobacteria, Acidobacteria, Actinobacteria, Bacteroidetes, Chloroflexi, and Verrucomicrobia). Proteobacteria was the core flora of the Korean pine root tips across all samples, regardless of location, age of the Korean pine forests, or date sampled. SpH, SEN, and SOM were the most significant factors that influenced the bacterial community of the rhizospheric soil. Soil origin and the plant itself were shown to be important factors driving the composition of bacterial communities in the Korean pine root tips. The interactions between bacterial families varied across different ecological niches based on the RLR and Spearman correlation analyses of the biomarker families in the root with bacterial species composition. The RLR of chemoheterotrophic bacterial functions predicted in the root tips was higher than the predictions for phototrophic and nitrogen functions. The impact of microbiome composition on the growth and health of Korean pine or other prized and ancient fossil ectomycorrhizal trees and specific ectomycorrhizal fungi, including mycorrhizal helper bacteria that could help mycorrhizal fungi navigate toward the root tips and enter the root to absorb nutrients, will be the subject of future studies.

### Funding

This work was supported by the Key Project on R&D of the Ministry of Science and Technology (No. 2018YFE0107800), and by the Chinese National Natural Science Foundation of China (No. 31600020). The funders had no role in study design, data collection and analysis, decision to publish, or preparation of the manuscript.

### Grant Disclosures

The following grant information was disclosed by the authors:
Key Project on R&D of the Ministry of Science and Technology: 2018YFE0107800.
Chinese National Natural Science Foundation of China: 31600020.

### Competing Interests

The authors declare there are no competing interests.

### Author Contributions

- Rui-Qing Ji and Shu-Yan Liu conceived and designed the experiments, authored or reviewed drafts of the paper, and approved the final draft.

- Meng-Le Xie performed the experiments, analyzed the data, prepared figures and/or tables, and approved the final draft.
- Guan-Lin Li performed the experiments, prepared figures and/or tables, and approved the final draft.
- Yang Xu analyzed the data, prepared figures and/or tables, and approved the final draft.
- Ting-Ting Gao performed the experiments, analyzed the data, authored or reviewed drafts of the paper, contributed materials, and approved the final draft.
- Peng-Jie Xing analyzed the data, authored or reviewed drafts of the paper, and approved the final draft.
- Li-Peng Meng conceived and designed the experiments, analyzed the data, prepared figures and/or tables, and approved the final draft.

## DNA Deposition

The following information was supplied regarding the deposition of DNA sequences:

The sequence data are available at NCBI SRA. The bacterial composition of root tips: SRR11350517, SRR11350518, SRR11350519, SRR11350520, SRR11350521, SRR11350522, SRR11350523, SRR11350524, SRR11350525 and SRR11350526. The rhizospheric soil bacterial composition: SRR11356286, SRR11356287, SRR11356288, SRR11356289, SRR11356290, SRR11356291, SRR11356292, SRR11356293, SRR11356294 and SRR11356295.

## Data Availability

The raw data is available in the Supplementary Files.

## Supplemental Information

Supplemental information for this article can be found online at http://dx.doi.org/10.7717/peerj.12978#supplemental-information.

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
