# Peer review of "Response of bacterial community structure to different ecological niches and their functions in Korean pine forests"

_PeerJ, doi:10.7717/peerj.12978_

## Round 0.1 · original submission · Major Revisions

· Academic Editor

Major Revisions

Dear Dr Li and Dr. Ji,

First of all please accept my apologies for the late evaluation. It was difficult to find competent reviewers but in the end, we received three evaluations. They give you helpful indications for your revision.

On average, the reviewers suggest major revisions for your paper.
One of the reviewers is more critical about your experimental set-up.

Please consider carefully the possible statistical problems arising from the selected experimental design.

Please consider all the indications you received from the reviewers, although you are not obliged to follow all the indications.

Sincerely,

Leonardo Montagnani

Reviewer 1 ·

Basic reporting

The authors studied how the bacterial community change in soil and roots of Korean pine in pure forests at different ages. They wanted to related community composition/structure and putative functionality to climate and soil chemistry. The introduction focus on different topics all important to understand why the authors choose to study this particular topic and which aspects are known and which are instead the open questions. I appreciate that the authors clearly state their objectives at the end of the introduction. In the material and method, authors should describe better how many samples were processed at the end as they state multiple cores were sampled but then pooled together.
Finally, the abstract need major revision, compared to the main text, it is difficult to read and do not provide the right information, too many sentence for results and no take home message.

Experimental design

In results: table are presented with average and SD but the authors should also include the number of samples used for this (n value= XXX). Table 1 authors state correctly that two sampling have the higher richness and diversity however this phrase is not supported by any statistic analysis please include. In section 3.1.2 it is not clear if the figure 2 include only data from CS1 or all the sampling points, while in the following part of the section authors analysed the data dividing them in CS1 alone and all the samples together, please clarify. Addtionally for table 1b CS1 does included only August samples or the total? And if the latter how the authors have taken in account the difference in number of samples. In general for alpha diversity data authors do not support their statement with statistical analysis. This trend is also continued in section 3.2.1 e.g. line 315/316 significantly higher but not stats are provided.
Figure 4, 5 should be redone. Relative abundance data are better presented as stacked histogram, the present figures are hard to ready and understand.

Validity of the findings

Discussion authors state that they did not found clear evidence of climate influence on the bacterial community however the difference in temperature is quite limited as all the sampling points are relative close, if winter sampling was performed a more marked difference in temperature would be found and maybe the results would have been different. Authors should state the difference in climatic variables are quite small and this could be the cause of the lack of effect. Line 450-451 how the authors can justify this phrase? Please explain better. Discussion is quite short and does not provide any additional information compared to the results, some interesting points are just briefly discussed and should be better explored. At the moment there is a clear lack of development of this section.

Additional comments

Sometime the text is repetitive or over wordy and can be simplified for an easier reading. E.g. line 102 Recently,…. In recent years.
Line 154 sapling -> sampling?

Reviewer 2 ·

Basic reporting

Understanding the composition, diversity and functions of bacterial community is a long-standing topic in ecology, and the authors provide a detailed study on these respects of bacterial community in Korean pine forests. Experimental design provides an intuitive perspective from which the complete dynamics of bacterial community can be obtained. Results are reasonable.

Experimental design

1. Why the authors do not show the soil bacterial and physicochemical properties of the blank plots (i.e. plots without planting trees)?
2. Not only the RDA, but also other models can be used for evaluating the relationship between bacterial community structures and environmental factors. Please to explain why the RDA method is used here for your purpose.
3. Line 160-165: There are reference for collecting rhizosphere soil samples of forests. Please to supplement relevant references here.
4. Line 167: Does “ice” mean the dry ice here? If not, it will affect the subsequent extraction of soil or plant bacteria, and the authors should ensure the accuracy of cryopreservation methods.

Validity of the findings

Overall, the research results are reliable, and the research conclusions are somewhat innovative.

Additional comments

INTRODUCTION
1. The objectives of this study seem too much and scattered, and it is suggested to further refine the research objectives.
RESULTS
2. Line 277-278: It can be seen from Table 1 (b) that CS1-G has the lowest Shannon index (2.73 ± 0.01). Please consider correcting the statement here.
3. Line 314-320: Authors should consider selecting the dominant bacterial taxa as the biomarkers of the different forest ages according to their feature importance.
4. Line 330-331: The five differently aged forests should be five different forest sites.
5. Line 378-395: Please to consider showing a table or figure to illustrate the relationship between climatic factors and soil physical and chemical properties separately, because of that the readers can see the result more visually.
6. In Figure 9 (a): RDA1 and RDA2 account for a small proportion, why?
DISCUSSION
7. The authors should further strengthen the analysis of the relationship between bacterial community functions and dominant bacterial taxa in different forest ages, and analyzed the difference of dominant bacterial groups in soils of different ages.

Reviewer 3 ·

Basic reporting

The authors aim to analyse the bacterial community structure of rhizosphere and root endosphere of Korean pine trees. They want to reveal differences in response to forest age and seasonal changes. However, the manuscript suffers from several shortcomings. I understand the intention of the authors, but the paper is exhausting to read and often formulated contradictorily.
The cited literature partly do not fit/prove to the statements and the data analysis seems not to be up to date.

Experimental design

My main concern is about the experimental design. The authors compared five forest plots but they did not use any replicates. Moreover, there are several other inconsistencies in the data analysis (see below).

Validity of the findings

Based on the experimental design and the inconsistencies in the data analysis (see below), the meaningfulness of the findings is doubtful.
I also do not think that the study is currently able to answer the four hypotheses.

Additional comments

- l. 64. Clarify the hypothesis and move to the other ones. The fourth hypothesis (l. 120) is incomprehensible.
- l. 50, 106. The references do not prove the given statements.
- l. 124. Please rephrase, clarify and shorten the whole section 2.1
- l. 166 “Composite…” Why did the authors not analyse the four plots separately??
- Please specify sampling of the rhizosphere.
- l. 186. The chosen primers do not exclude cp and mt 16S rRNA genes. What about the percentage in the root samples?
- The authors should think about using a ASVs pipeline instead of the traditional OTU approach. Further, they applied outdated databases (e.g. RDP release 11 instead of the current release 18). A confidence threshold of 0.5 is very unusual. Why did the authors use this low confidence value (in contrast to 80%)? This influences decisively the community structure especially at the genus level.
- l. 212 The subsampling should be specified in Material and Methods (not in the Results).
- Difference between Bray-Curtis and UniFrac should be revised and clarified.
- “p < 0.01 was considered extremely statistically significant.” Please rephrase.
- The first paragraph of the Results section is very confusing. First, I do not understand “we generated 20 gross samples from 1,000 subsamples”. Then you provided the total number of sequences (420,000) and in the next sentence again with 690,000… Further, following your subsampling procedure (minimum 26,088/0.75 * 20), you should have much more sequence reads. And the last sentence seems not to make any sense in this context.
- In the next paragraph, the authors mentioned results of a PCA but in M&M a PCoA is described. Further, it should be revised to show a result not a list of sampling points. The differences should be tested for significance by ANOSIM, MRPP or a similar procedure. Please also mention if there was an impact of plant age and the season on the community structure. I understand that this was your main topic.
- The alpha diversity indices were provided with a SD. However, the authors did not study replicates!
- l. 532 The given pipeline in M&M does not allow the assignment at the species level.
I want to stop here as I think it already suggests that the complete story should be clarified and revised.

---

## Round 0.2 · Major Revisions

· Academic Editor

Major Revisions

Dear Dr Ji and Dr Xie,

We received two evaluations of your revised paper. While one reviewer was satisfied by your answers and changes in the text, the second one found relevant problems in the statistical analysis you performed.
Your paper will be fully considered again only once these indications will be fully addressed.

Sincerely,

Leonardo Montagnani

Reviewer 2 ·

Basic reporting

The author has carefully revised the all sections according to the Reviewers’ suggestions. Taken altogether, the paper has met the basic requirements for publication. It is suggested that this MS need further language editing before the publication.

Experimental design

The authors have carefully revised the Materials and Methods part according to suggestions of the reviewer, in particular the Study sites and experimental design, and Sampling and processing of the root tips and rhizospheric soil.

Validity of the findings

This MS analyzed the composition of bacterial communities and their affecting factors in the rhizospheric soil and root tips of Korean pine forests with different ages, which is helpful to improve soil nutrient management in artificial Korean pine forests.

Reviewer 3 ·

Basic reporting

no comment

Experimental design

no comment

Validity of the findings

First, I want to reply to the answers of the authors:
I understand and appreciate that the authors studied five different forest plots. However, I disagree that the two similar plots can be used as replicates. The basis of any statistical analysis is the comparison of within and between-group variability. In view of this, the four sample areas per plot should not have been pooled! It’s a pity and limits the meaningfulness of the paper. As a result, you have only one observation per plot and you cannot analyze statistical differences for the alpha or beta diversity (Table 1 without SD and statement of significance; pCoA but without a statement of significance, e.g. using PERMANOVA or ANOSIM). Please be careful with conclusions for the bacterial community for the different forest plots. The respective statements in Abstract, Results and Discussion should be revised.

Further, Figure 1-3 just show the difference between root and rhizosphere microbiome. These differences between both habitats are well known and have been studied for several years.
The results shown in Fig. 4, 5 and Table 1 do not substantiate differences between the forest plots and fit to the pCoA ordination plot. There is no clear difference and without any knowledge about the variation within the plots, the authors cannot conclude unambiguous results. (BTW, please avoid the word “significantly” in context with the 5 forest plots, as you did not analyze significant differences.)
Figure 6 contains error bars. Without any replicates, I wonder about the basis for the calculation.

All in all, I suggest a complete revision as stated above.

---

## Round 0.3 · Minor Revisions

· Academic Editor

Minor Revisions

Dear Dr. Ji and Dr. Xie,

I checked again your manuscript. While the scientific aspect is satisfying now, I note that there are several grammar errors throughout the text.

I recommend professional editing of your manuscript before final acceptance.

Sincerely,

Leonardo Montagnani

---

## Round 0.4 · accepted · Accept

· Academic Editor

Accept

Dear Dr Ji and Dr Xie,

I am pleased to inform you that I consider your paper acceptable now. Thank you for considering Peerj for publication.

Sincerely

Leonardo Montagnani